# Overview of Quantitative Methodologies to Understand Antimicrobial Resistance via Minimum Inhibitory Concentration

**DOI:** 10.3390/ani10081405

**Published:** 2020-08-12

**Authors:** Alec Michael, Todd Kelman, Maurice Pitesky

**Affiliations:** 1Department of Population Health and Reproduction, School of Veterinary Medicine, UC Davis, 1089 Veterinary Medicine Dr., VM3B, Davis, CA 95616, USA; tjkelman@ucdavis.edu; 2Department of Population Health and Reproduction, School of Veterinary Medicine-Cooperative Extension, UC Davis, 1089 Veterinary Medicine Dr., VM3B, Davis, CA 95616, USA; mepitesky@ucdavis.edu

**Keywords:** MIC, AMR, mixed effect models, logistic regression, cumulative logistic regression, accelerated failure time-frailty models, mixture models

## Abstract

**Simple Summary:**

An emerging threat to human and food animal health is the development of antimicrobial resistance in bacteria associated with food animals. One of the primary tools for assessing resistance levels and monitoring for changes in expressed resistance is the use of minimum inhibitory concentration tests, which expose bacterial isolates to a series of dilutions of an antimicrobial agent to identify the lowest concentration of the antimicrobial that effectively prevents bacterial growth. These tests produce a minimum inhibitory value that falls within a range of concentrations instead of an exact value, a process known as censoring. Analysis of censored data is complex and careful consideration of methods of analysis is necessary. The use of regression methods such as logistic regression that divide the data into two or three categories is relatively easy to implement but may not detect important changes in the distributions of data that occur within the categories. Models that do not simplify the data may be more complex but may detect potentially relevant changes missed when the data is categorized. As a result, the analysis of minimum inhibitory concentration data requires careful consideration to identify the appropriate model for the purpose of the study.

**Abstract:**

The development of antimicrobial resistance (AMR) represents a significant threat to humans and food animals. The use of antimicrobials in human and veterinary medicine may select for resistant bacteria, resulting in increased levels of AMR in these populations. As the threat presented by AMR increases, it becomes critically important to find methods for effectively interpreting minimum inhibitory concentration (MIC) tests. Currently, a wide array of techniques for analyzing these data can be found in the literature, but few guidelines for choosing among them exist. Here, we examine several quantitative techniques for analyzing the results of MIC tests and discuss and summarize various ways to model MIC data. The goal of this review is to propose important considerations for appropriate model selection given the purpose and context of the study. Approaches reviewed include mixture models, logistic regression, cumulative logistic regression, and accelerated failure time–frailty models. Important considerations in model selection include the objective of the study (e.g., modeling MIC creep vs. clinical resistance), degree of censoring in the data (e.g., heavily left/right censored vs. primarily interval censored), and consistency of testing parameters (e.g., same range of concentrations tested for a given antibiotic).

## 1. Introduction

The World Health Organization has deemed antimicrobial resistance (AMR) one of the most urgent health threats of our time [1]. Antimicrobial-resistant bacteria are implicated in approximately 2.8 million cases of clinical infections and an additional 35,000 human deaths annually [2]. In addition to clinical burdens, the estimated annual economic cost of AMR is approximately $3 trillion in GDP loss due to excess health care and loss of productivity [3]. The issue of AMR affects humans, animals, and the environment alike, prompting the World Health Organization to recommend the coordinated monitoring of resistance in bacteria from food, food animals, and clinically ill patients to understand the patterns of AMR and how it affects the entire food production chain [4]. As a part of this effort, the ability to monitor AMR phenotypically and genotypically with respect to resistance is an important methodological tool in public health and food production [5].

To address the threat of AMR, national and international surveillance systems conduct widespread antimicrobial susceptibility testing to track resistance levels in microbial populations. The U.S. National Antimicrobial Resistance Monitoring System (NARMS) assesses resistance levels of enteric bacterial isolates from retail meats, livestock, and humans [5]. For this purpose, surveillance programs measure resistance levels of isolates to each antimicrobial using minimum inhibitory concentrations (MIC) of selected antibiotics. The MIC quantifies the lowest tested concentration at which an isolate’s growth is inhibited by a specific antimicrobial [6]. The MIC is determined for a given isolate and antimicrobial using automated instrument systems or manual testing methods, including agar dilutions, antimicrobial gradient method (e.g., E-tests), and broth dilutions [7,8]. These tests expose the isolate to a series of concentrations of antimicrobials on a two-fold scale, and the lowest concentration where inhibition of bacterial growth is present is reported as the MIC for that isolate and the corresponding antimicrobial compound.

As a result of this method of data collection, the single MIC value reported represents an interval that contains the exact MIC where inhibition was observed on a log2 scale (i.e., the true MIC lies between the smallest concentration tested where inhibition was observed and the highest concentration tested where inhibition was not observed). Due to this testing structure and the limits on the ranges of concentrations of antimicrobials tested, MIC data is subjected to three types of censoring: left, right, and interval. The term “right-censored” refers to an observation where the MIC is greater than the highest antimicrobial concentration tested (i.e., there was no inhibition of growth observed) and is reported as “>J μg/mL” where J is the highest concentration tested. In contrast, “left-censored” refers to an observation known only to be below the lowest antimicrobial concentration tested (i.e., there was inhibition of growth at all dilutions) and is reported as “≤J μg/mL” where J is the lowest concentration tested. “Interval-censoring” refers to observations that are known to lie between two values and is the typical case with MIC data when the range of concentrations tested captures the distribution of MICs well. In the case of MIC data, interval-censoring means the true MIC lies between the reported MIC and the value one step below on the two-fold scale (e.g., between 2 and 4 µg/mL). If the reported MIC value is K, then the interval of possible concentrations for inhibition would be $\left(2^{\left({log}_{2}K\right)-1},K\right)$. As an example, if the growth of a bacterial isolate is inhibited at a concentration of 32 μg/mL of Streptomycin and the isolate is able to grow at a concentration of 16 μg/mL, then the MIC is reported as 32 μg/mL, and the true MIC lies in the interval between 16 and 32 μg/mL.

These MIC tests are typically conducted in vitro on planktonic bacteria. However, it is important to note that resistance levels are not fixed and may change based upon the environment and the biological state of the bacteria (e.g., planktonic, biofilm, bacterial spore), which can impact bacterial resistance levels through adaptive resistance, as opposed to acquired resistance [9,10]. While there are methods for testing the AMR levels of biofilms, it is important to recognize that not all conditions can be accounted for during in vitro MIC testing [11,12].

DNA sequencing and associated molecular techniques such as whole genome sequencing (WGS) and metagenomics offer a parallel approach to phenotypic susceptibility testing with some inherent advantages from collection of genomic data as a method for the identification of AMR based genes. Using genomic approaches it is possible to accurately estimate the phenotypic resistance of isolates using genotypic information derived from WGS in some situations [13,14]. For example, high correlations between resistance genes identified in *Campylobacter* spp. and phenotypic resistance were observed for a panel of antimicrobials, and machine learning techniques applied to WGS have been able to accurately predict MIC results for *Salmonella* isolates [15,16].

Based on these types of results, WGS data is understood to play an important role in identifying resistance patterns and can be used to supplement phenotypic information in surveillance programs as well. However, while genotypic methods have proved effective predictors of phenotypic resistance in some studies, they may suffer when used for many bacterial species where the genetic resistance profile is incomplete [17,18]. As a result, it is important to recognize that there are still important functions for phenotypic-based analyses such as MIC. Specifically, phenotypic methods are essential for verifying that resistance levels have not significantly deviated relative to previous levels, and MIC data remains an important tool to allow for a better understanding of genomic data and how it relates to resistance [19]. Additionally, in areas where surveillance programs lack widespread access to large-scale sequencing, phenotypic analysis provides important practical information on trends in antimicrobial resistance [20]. In summary, there are advantages and disadvantages of genotypic and phenotypic methods and it is important to further optimize the methods and analysis techniques of both approaches. To that point, this article is intended to offer a review of the litany of data handling approaches and corresponding regression techniques.

## 2. Regression for Dichotomized Minimum Inhibitory Concentration (MIC) Data

### 2.1. Epidemiological Cutoffs and Clinical Breakpoints

There are a number of different ways that MIC data is modified for use as the dependent variable in regression and analysis. In general, different modeling approaches utilize data handling methods that balance model simplicity with information loss. Logistic regression requires a dichotomous outcome, and the breakpoints are carefully selected to divide observations into separate meaningful outcome categories.

One method for selecting this dividing value on the MIC scale is to use the epidemiological cutoff value, which separates the distribution of MICs of wild type (WT) isolates, which lack phenotypically-detectable acquired resistance mechanisms, from the non-wild type (non-WT) organisms that possess phenotypically-detectable acquired resistance mechanisms. [18,21]. The European Committee on Antimicrobial Susceptibility Testing (EUCAST) sets epidemiological cutoff values (ECOFF) based on the consensus of visual and mathematical analyses of observed MIC distributions performed by identifying WT and non-WT distributions in the bimodal distribution of MICs [22,23]. The mathematical determination of the ECOFF for each antimicrobial agent is done using a tool called the ECOFFinder, which fits a cumulative log normal curve to the WT MICs and identifies cutoffs that classify 97.5% and 99% of the WT isolates correctly [24]. A similar method is employed by the Clinical and Laboratory Standards Institute (CLSI) for its own epidemiological cutoff values (ECV) [22,25].

The other predominant categorization method involves the establishment of clinical breakpoints. This process partitions MIC values into distinct classes of bacterial susceptibility based on clinical outcomes. CSLI defines these classes as “susceptible” (S), “resistant” (R), and either “intermediate” (I) or “susceptible-dose dependent” (SDD) [26]. Note that EUCAST defines its intermediate breakpoint as “susceptible, increased exposure” (I) [27]. See Figure 1 for a visual comparison of MIC concentrations, censored MIC intervals, clinical breakpoints, and epidemiological cutoffs. [27,28]. These breakpoints are established by organizations such as the CLSI, EUCAST, and the U.S. Food and Drug Administration Center for Drug Evaluation and Research [28]. Different organizations use different names and definitions for intermediate classes that allow for some ambiguity in interpretation and specific information on these categories is typically available on each organization’s website [27,28]. Classifications based on an ECOFF are not immediately related to classifications based on clinical breakpoints, an isolate identified as non-WT will not necessarily be clinically resistant, instead it may still be clinically susceptible [18].

The clinical breakpoints and ECOFF seek to avoid splitting the WT isolates into multiple categories to avoid accidental classification of a WT isolate to a higher resistance category. This approach must account for about one dilution of variation in measurement introduced by testing procedures in breakpoint or cutoff determination, as well as inter-laboratory variance [8,22,29]. As a result, the ECOFF must account for this variation as it attempts to separate this distribution of isolates from the non-WT bacteria that are higher in the range of tested concentrations.

Though this approach simplifies the modeling approaches necessary to look for trends and increases the overall ease of interpretation, a portion of the information contained by the MIC value is lost when it is dichotomized or categorized, which will be further discussed later in the review [30,31,32]. Additionally, breakpoints may be subject to change over time in response to changes in protocols, resistance patterns, and guidelines [33].

### 2.2. Logistic Regression

The most common regression approach for MIC data is to use dichotomized data as the outcome for logistic regression, which identifies effects of an exposure or time (e.g., source animal or year of sample collection) on resistance using a generalized linear model with a logit link [34]. Logistic regression is a flexible method that allows for investigation of a temporal trend or exposure of interest while adjusting for the effects of other relevant covariates, with the capacity to include random effects. Additionally, logistic regression facilitates the calculation of odds ratios and confidence intervals for the temporal trend or exposure [35,36]. Logistic regression requires a dichotomous outcome, so clinical breakpoints with an intermediate category must be treated according to the focus of the study and the definitions of I to reduce the number of levels to two. For instance, a study interested in the clinical aspects of resistance would collapse the S and I categories together, because EUCAST defines I as having a high probability of therapeutic success if adjustments are made to dosage or the drug is concentrated at the site of infection, similar to the SDD category described by the CLSI [37].

Logistic regression can account for correlated samples (e.g., as a function of a specific laboratory’s process) by inclusion of random effects (e.g., for each laboratory) in a generalized linear mixed model [34].

Logistic regression has been used in the literature to explore the effects of many variables on levels of resistance, examples include a test for temporal trends in tetracycline resistance among *E**scherichia coli* isolates from pigs in Aerts et al. [38]. In their example model, dichotomized MIC data is used as the outcome in a logistic regression with time as the independent variable to test for significant temporal trends over several years [38]. Another study examining resistance to *E. coli* among dairy cattle use several multivariable logistic regressions to explore resistance to tetracycline, ampicillin, and several other antimicrobials separately, with final models including some of the following covariates: herd size, herd parity, barn type, average herd milk production, region of dairy farm, and separate variables for use of different antimicrobials [39]. 

### 2.3. Considerations

Logistic regression has been an established and accepted method of MIC data analysis for some time, and it has seen a variety of uses, including for surveillance programs such as the European Centers for Disease Control and Prevention’s exploration of antimicrobial consumption and AMR and the examination of temporal trends in NARMS clinical data [40,41]. The benefits of logistic regression include its familiarity to many researchers in the field of AMR and the ease of interpretability for its results and methods [42]. As discussed previously, logistic regression can be much more insightful than descriptive statistics for the investigation of AMR, as it can adjust for covariates (e.g., source species, husbandry practices) before examining temporal trends and risk factors while accounting for the impact of study design to minimize the risk of underestimating standard errors [43,44,45]. Logistic regression may also be a useful method when the MIC distributions are heavily left and right-censored, as it is relatively unaffected by the lack of information on the shape of the distribution of the WT and non-WT groups [38]. Also, logistic regression is able to analyze data from laboratories that test different ranges of MICs for a given antibiotic and organism, as it collapses all MIC levels above and below the cutoff into WT and non-WT groups which then may be compared across laboratories.

### 2.4. Information Loss and Minimum Inhibitory Concentration (MIC) Creep

Information loss is a byproduct of the dichotomization of MIC data, as interval-censored, continuous data is simplified to being either above or below a threshold value (see ECOFF, ECV, and clinical breakpoints above), which results in data loss [30,31]. With dichotomized MIC data, only changes in the proportions of MIC values that fall on each side of the threshold are captured via logistic regression. In other words, only fluctuations in the proportions of WT and non-WT, or S and R, are captured by logistic regression, and all fluctuations of values within each category are lost. As a result, surveillance of dichotomized AMR data can result in the loss of critical information such as changes in trends in the mean MICs of the S and R, or WT and non-WT, groups if those changes do not cross threshold values. This phenomenon where changes in the mean MIC are not detected by dichotomized analysis of MIC data is referred to as “MIC creep” and “MIC decline.” In response to information loss through dichotomization, a wide array of methods of analysis incorporating additional information from MIC data have been developed. These methods examine the entire scale of MICs to gain insight into the distributions of MICs above and below the cutoff values [38]. As a practical example, MIC creep has become a relevant concern for surveillance of increasing resistance to vancomycin among methicillin-resistant *Staphylococcus aureus*, though individual study results differ with regards to the nature of this trend [46,47,48].

In another study, two nonparametric approaches and logistic regression were used on subsets of NARMS retail and slaughter data to demonstrate significant changes in the distributions of MICs from consecutive years. MIC creep was identified via the nonparametric tests but not detected by logistic regression. In contrast to logistic regression, the nonparametric approaches treat each interval as a separate unordered category, then evaluate whether the distribution of MICs among these categories changes between two time points either a year apart or between slaughter and retail [30]. The results of the study suggest that the distributions of MIC data may vary without being detected by methods that dichotomize the data. Therefore, these data indicate that the entire scale of values needs to be incorporated in the analysis of MIC data [30].

In the interest of retaining more of the information lost through dichotomization, there are many modeling approaches for MIC data that do not dichotomize the data, several of which will be discussed later in this review. An alternative proposed approach allowing for mixed linear regression is to ignore censoring of the MIC data and instead treat MICs as observations on a continuous scale. However, this interpretation leads to an overestimation of the means and underestimation of the standard errors, increasing the likelihood of type I error [49,50].

## 3. Models for Ordinal Data

### 3.1. Cumulative Logistic Regression

In order to address interval censored MIC data, the intervals can be treated as ordinal data, which preserves the natural order of the MIC intervals while treating each MIC value as a category. In contrast to the dichotomization with standard logistic regression, treating MIC data in an ordinal fashion preserves the ordered relationship of the MIC intervals without collapsing the categories. In this approach, the intervals are regarded as a series of ordered categories, although ordinal data does not retain the two-fold relationship between intervals on the MIC scale, meaning the numerical relationship between the categories is lost but rank is retained. Two types of cumulative logistic regression models may be employed for analysis of MIC data: the proportional odds cumulative logit model and the generalized cumulative logistic regression model [38,51,52]. For the purposes of regression, the proportional odds cumulative logit model [53] is as follows:(1)$$\mathrm{logit}\;[P(y\leq j)]=\alpha_{j}+\beta \;x$$
This model adapts logistic regression to suit ordinal data, where the effects of time or other exposures on cumulative odds ratios can be explored. The model uses the logit of cumulative probabilities, which describe the probability that an observation y will fall into a category j or any of the categories below it on the ordinal scale, expressed as P(y≤j) where there are a total of J categories [38,53].

For the proportional odds cumulative logit model, it is important to check the validity of the proportional odds assumption before using this approach. Specifically, the odds ratio of an effect for every possible dichotomization of the J categories should remain the same. In other words, the coefficient for the effect of each variable x should remain fixed for each possible split of the groups into two (i.e., into “category j or below” versus “above category j” for categories 1 through J-1). For instance, the odds ratio for the effect of x is the same for the logistic regression with the two outcomes y ≤ j and y > j and the logistic regression with the outcomes y ≤ j + 1 and y > j + 1, such that the coefficient for x in each comparison is the same, though the intercept α_j_ for each outcome level j may vary [54]. It may be helpful to think of this as performing simultaneous logistic regressions for each of the possible separating points with the same coefficients for the independent variables, but different intercepts. The proportional odds assumption may be violated in some instances by MIC data, and as such should be carefully examined before employing this approach [55]. When the proportional odds assumption is not met, there are alternatives such as generalized ordered logit models, which allows for ordinal regression when the proportional odds assumption is not met, and multinomial regression models, which disregard the order of the categories [52,54]. The generalized ordered logit model allows the coefficients, and therefore the odds ratios, to vary for each value of j in addition to the intercepts, making this model considerably more flexible than the proportional odds cumulative logit model [54]. The tradeoff of this approach is an increase in the number of parameters required to fit the model, which may lower bias at the cost of increased variance and a risk of overfitting [53].

### 3.2. Applications of Regression Approaches for Ordinal Data

Ordinal regression methods using MIC as an outcome are rare in the literature. The usefulness of the proportional odds cumulative logit model is primarily limited to analysis of suspected progressive drift in MICs, an example of which can be found in Catania et al. [56]. In this study, the proportional odds cumulative logit model was used to test for temporal trends in tilmicosin MIC, which were hypothesized to exhibit gradual increase towards resistance (i.e., MIC creep). The model included the covariates “year” and the genotype of the *Mycoplasma synoviae* isolate, along with an interaction term [56]. Similarly, a second application of the proportional odds cumulative logit model can be found in Bote et al. [57] which examines two narrow distributions of primarily susceptible MICs from *E. coli* isolates challenged with glyphosate isopropylamine salt and glyphosate-containing formulation. The two regression models for this study, one for each herbicide, used the following covariates: “time of isolation” of the bacteria, whether the isolate was commensal or pathogenic, extended spectrum beta-lactamase (ESBL) producing strains and non-ESBL producing strains, and source species [57].

Other studies have used MIC data as an ordinal outcome for a model relating genotypes to the phenotypic outcome or to examine the effect of both year and genotype on phenotype [56,58]. Additionally, although not being used to directly estimate MIC values, the generalized ordered logistic regression model has been applied to multi-drug resistance (MDR) data, where the approach was used to model the association between chlortetracycline (CTC) and copper use and MDR in *E. coli* isolates [52]. The analysis used the number of classes of antimicrobials to which an isolate possessed phenotypic resistance as the outcome and the regression covariates included: copper and (CTC) usage along with time, classified into pre-treatment, during treatment, and post-treatment. The phenotypic resistance to each class of antimicrobial agents was determined based on dichotomous breakpoints for each antimicrobial. The results showed that the proportionality assumption for ordinal logistic regression was violated, necessitating the use of partially constrained generalized ordered logistic regression, where some of the coefficients remained the same for all values of j, while others vary between levels on the ordinal scale [52]. Additionally, another study used cumulative logistic regression to analyze multidrug resistance among *E. coli* isolates from dairy calves, where the outcome of interest was 24 ordered clusters of MDR patterns based on the MIC data. Ordinal regression was used to examine relationships between the outcome and the covariates: time, antimicrobial usage, and whether the antimicrobial use was therapeutic or prophylactic [59]. 

### 3.3. Considerations for Cumulative Logistic Regression

The primary benefit of ordinal logit models is that they reduce the amount of information loss relative to logistic regression by avoiding reductions in the number of categories and maintaining the order of these categories. In addition, maintaining the data as categorical avoids the need to fit a distribution to the data and reliance on a single cutoff point is avoided. However the analyses would not be applicable if the range of MIC values tested change for a given organism and antimicrobial agent, because of mismatched ordinal categories in the outcome [38]. As such, these methods would be ill-suited for data sets gathered from sources using different ranges of concentrations for MIC testing. Additionally, the lack of a specific numerical relationship between ordinal categories makes other models better suited for finding the properties of the underlying continuous distribution or distributions. Finally, the scarcity of ordinal regression techniques in the literature may make communications of methods and findings more difficult, though the similarities between these methods and logistic regression in approach and interpretation may make ordinal regression more accessible to non-statisticians than more complex approaches.

## 4. Models on the Continuous Scale for Interval-Censored Data

Avoiding information loss from statistical models that treat MIC data as categories requires methods that operate on the continuous scale but do not ignore censoring. Here, we present mixture models and accelerated failure-time models for MIC analysis that model MICs on the continuous scale while accounting for their interval-censored nature.

### 4.1. Mixture Models

One approach is the use of a mixture model (not to be confused with a mixed model) that treats the population of bacteria tested as a collection of multiple subpopulations. These subpopulations are distinguished by the presence of resistance mechanisms that are mixed into a bimodal distribution consisting of WT and non-WT components [38]. Unlike logistic regression models, these models need not rely on a cutoff to classify observations, and some implementations use latent variables to account for the interval-censored data and mixing weights to estimate the prevalence of WT and non-WT isolates [60,61].

Mixture models are useful for models made of components that follow different distributions. This approach is useful in this instance as MIC data can form a bimodal distribution consisting of WT and non-WT isolates. In these distributions, the WT component forms the lower peak of the bimodal distribution and is composed of the bacteria without acquired resistance mechanisms with some variation around the mode primarily introduced by methodological differences [62]. In contrast, the non-WT component represents the MICs of isolates with phenotypically-detectable acquired resistance mechanisms that form the peak of the bimodal distribution at higher MIC values, with the variance of this component attributed to differences in resistance mechanisms as well as variation introduced during MIC measurement [29,62]. Additionally, studies have shown that the non-WT component may itself be multimodal and use a mixture model where the non-WT component is composed of many overlapping non-WT components [63,64].

Turnidge et al. found the WT population’s log_2_ MIC distribution fits a log-normal distribution, though the cumulative log-normal distribution may be a better fit as it is less affected by the grouping of the data [23]. The study reached this conclusion using nonlinear least squares regression to fit cumulative counts of log_2_ MIC frequencies to cumulative normal distributions while also approximating the number of bacteria that were part of the WT component of the bimodal distribution [23]. A later study used the multinomial distribution and the Akaike information criterion to test the log-normal and gamma distributions and argues that averaging these two fitted distributions may provide a better overall fit for the WT component [62].

To fit a continuous distribution to MIC data assumed to contain multiple subpopulations, Jaspers et al. employed a semi-parametric mixture model [62,65]:(2)$$f\left(x\right)={\pi \;f}_{1}(x\;|{\;\theta }_{1})+\left(1-\pi \right){\;f}_{2}(x\;|{\;\theta }_{2})$$
This model employed a non-parametric approach for the multimodal non-WT component ($f_{2}$) and a parametric form for the WT component ($f_{1}$) which was assumed to follow a fixed log-normal distribution (i.e., the WT population is presumed not to change). In this mixture model, the $\theta_{i}$ represent the parameters of the component distributions, x represents the MIC interval, and π represents the mixing weight. This version of the model did not include covariates and employed curve-fitting to estimate the density distributions of these components. Using this approach, the first component can then be used to determine a cutoff between the WT and non-WT distributions, much like the ECOFF. Note that in this case, the mixing weight serves as an estimate of WT prevalence.

The model was built by first fitting a log-normal distribution to the WT component, then multiple log-normal distributions were fitted to compose the multimodal non-WT distribution ($f_{2}$) [62]:(3)$$f_{2}(x\;|{\;\theta }_{2})=\overset{n}{\underset{i=1}{\sum }}\tau_{i}{\;f}_{i}(x\;|{\;\theta }_{2i})\;$$

The non-WT component may be thought of as a mixture of n log-normal distributions ($f_{i}$) with mixing weights of $\tau_{i}$, to capture the heterogeneous non-WT subpopulations arising from different acquired resistance mechanisms [63].

This study’s modeling approach primarily focused on the behavior of the non-WT bacteria, while other mixture modeling approaches may emphasize the WT bacteria, and their utility may depend on the degree of left and right-hand censoring in the model and the focus of the study or priorities of the surveillance program [63]. 

This method was later refined by Jaspers et al. to allow for simultaneous estimation of the WT and non-WT components, as they found estimating the WT component first resulted in inaccurate estimates of standard errors [64].

The iterations of the Jaspers et al. mixture model approach presented so far have been used to the fit a curve to the shape of the MIC data, as opposed to modeling the effects of covariates on prevalence of WT phenotypes. In the next iteration of the model, a Bayesian framework was used to estimate component densities and to allow the mixing weights ($\pi ,\;\tau_{i}$) of the components to vary for levels of covariates such as year (e.g., the mixing weight of the WT component may decrease from one year to the next to indicate a decrease in WT prevalence in that time) [66]. The parameters (θ) of the non-WT Gaussian distributions ($f_{i}$) in this model are not dependent on the covariates, so to allow for changes in the means of log-normal non-WT distributions ($f_{i}$), the number of distributions in the non-WT component is not constrained. This allows Gaussian distributions that previously had a mixing weight of zero to increase their mixing weight and replace another Gaussian distribution in place of changing the mean and standard deviation of the previous curve [66]. Though it is useful for observing changes in MIC distributions for surveillance purposes and can detect changes in proportions of bacteria with reduced susceptibility, this approach is not a regression technique and was not designed to establish causal relationships between covariates and trends in MIC distributions.

A later iteration of this Bayesian approach by Jaspers et al. enables the modeling of joint MIC distributions to explore relationships between resistance to multiple antimicrobials as a way to explore multidrug resistance patterns [67]. While this review will not cover Bayesian statistics extensively, the utility of a Bayesian framework for mixture distributions in this context is to take a prior estimate of the distribution and to update it using information from the observed MIC data and covariates to produce a posterior estimate of the parameters of the component distributions and the mixing weights [68].

Another study employed a type of hierarchical Bayesian mixture model that also presumes the MIC data to be composed of overlapping WT and non-WT distributions [60]. This model assumes both components follow a log-normal distribution with fixed standard deviation. Additionally, the model introduces a linear temporal trend for the log_2_ mean of the log-normal distribution of the WT MIC data, allowing for possible detection of MIC creep or decline in the WT group and also some estimation of the magnitude of effect of time on the log_2_ mean WT MIC. In contrast to the mixture models previously described in this review, this approach introduces regression by allowing the WT population mean to vary with the covariates (e.g., time) while keeping the non-WT population mean fixed, largely due to heavy amounts of right-censoring in the data sets employed in the study. Additionally, the mixture model estimates prevalence of non-WT isolates along with a variance for this value, with this calculation relying on the model’s ability to distinguish between the WT and non-WT distributions instead of using a single cutoff point such as the ECOFF [60]. It should be noted that this approach requires several assumptions including normality of the non-WT distribution, whereas the previous model assumed the non-WT to be a mixture of multiple normal distributions, and a fixed variance for the WT and non-WT normal distributions [60].

### 4.2. Considerations for Mixture Models

This approach to analyzing data accommodates interval-censored data, which prevents information loss caused by simplification of the MIC data. The fitted mixture model operates on a continuous scale, meaning it provides detailed estimates of the entire distribution of MICs. Estimating the underlying distribution of MICs could prove useful for identifying changes in these distributions that might provide earlier detection of resistance trends for surveillance programs [66]. Additionally, the classification of the model into components allows for estimates of resistance prevalence [63]. Mixture models used in parallel with other phenotypic and genotypic methods could be a useful surveillance-based approach for identifying changes in MIC distributions over time.

It should be noted that use of these mixture models requires the MIC data to be on a uniform scale. Therefore, the ranges of concentrations of antibiotics tested for a given organism must be identical for all observations, which could present difficulties when analyzing data collected from multiple sources. In addition, mixture models require a distinct separation to be present between the WT and non-WT components, which may be violated by some MIC distributions. It should also be noted that the classification performance of mixture models is impacted by the degree of separation of the two components, as greater separation improves the model’s ability to classify observations [66].

### 4.3. Accelerated Failure Time Models

In addition to mixture models, survival analysis methods are commonly used when working with interval-censored data, suggesting possible applications to MIC data. Multiple studies have explored applicability of the Cox proportional hazards model to MIC data, as it is often used with censored data in survival analysis. However, as with ordinal regression discussed previously, multiple studies have found that the assumption of a fixed hazard ratio over the range of concentrations tested (i.e., proportional hazards, as opposed to proportional odds for ordinal regression), was routinely violated by MIC data and the interval-censored nature of the data created further issues with this approach to regression [38,69].

A parametric survival analysis technique suited for working with interval-censored data is the accelerated failure time (AFT) frailty model, which is a parametric technique used to model time-to-event data that incorporates specific random effects called frailty components [38,70]. In the context of survival analysis, data are often presented as “time-to-event” where the dependent variable contains information on the amount of time before an event (e.g., relapse, death, part failure) occurred. When no event occurs within the time period of the study, the survival time is considered to be censored since the time to event is unknown. Similarly, MICs can be thought of as the increasing concentration of an antimicrobial that an isolate is exposed to before an event occurs, where the event is inhibition of growth. All three types of censoring present in MIC data can also occur in survival analysis under certain data collection methods and can be accommodated in AFT models. Thus, MIC data can be analyzed in this fashion using antimicrobial concentration as time and inhibition of growth as the event [69]. As the growth of an isolate is more likely to be inhibited at higher antimicrobial concentrations, concentration-to-inhibition of growth represents a monotonically increasing hazard [34]. AFT regression takes the following form [71]:(4)$$\;\log\left(T_{i}\right)=\beta_{0}+\beta_{p}x_{\mathrm{pi}}+\sigma \epsilon_{i}$$
This allows for the incorporation of covariates (x_pi_) and their effects (β_p_) into the model that in the context of MICs will increase or decrease the expected concentration (T_i_) to inhibition [72]. The distribution of T_i_ determines what distribution the error term (ε_i_) will follow to fit the AFT model to the data once the scale factor (σ) is included. Examples of possible distributions for T_i_ include the Weibull, exponential, or log-logistic model [71].

An example of the application of an AFT model for testing for temporal trends in MIC data can be found in Aerts et al., where the susceptible and resistant populations are modelled separately, requiring the use of a cutoff value. The regression found growing separation between the means of two groups is observed over the course of several years, with time as the exposure of interest included in the model [38].

Another study used NARMS data to compare the AFT frailty model to a mixed effects logistic regression by analyzing resistance to eleven antimicrobials with the fixed effects: year, source animal species, and an indicator of multidrug resistance; laboratory study site was included as a random effect [34]. For the AFT, a Weibull distribution was used for the hazard [34]. This is appropriate for models where a hazard is monotonically increasing or decreasing, as in the case of antimicrobial resistance testing [73]. In the study, both the AFT and logistic regression found the multidrug resistance indicator to be significant for all antimicrobials, but for some antimicrobials there was disagreement on the magnitude, though not direction of temporal trends. In addition, for one antibiotic (cephalothin), conflicting temporal trends were identified, with the logistic regression indicating an increasing trend while the AFT identified a minor decrease in resistance followed by an increase over time. Additionally, the AFT detected significant effects of source animal species for all eleven antimicrobials, while the logistic regression found that covariate to be significant in only nine. Additionally, the AFT often classified more of the species effects within each regression as significant relative to the logistic regression, suggesting a greater degree of sensitivity in the AFT model [34]. This example illustrates the critical importance of selection of modeling approach, as the type of regression used can have a large impact on the findings of the study.

One weakness of this approach was illustrated when the study found that the AFT model returned extreme values for model parameter estimates and between study-site variance associated with high levels of left and right censoring. This suggests that AFT models may not find an optimal solution when large numbers of MICs are outside the tested range or are not well distributed [34].

### 4.4. Considerations for Accelerated Failure Time Models

Using regression methods such as AFT that minimize information loss (i.e., does not categorize the MIC values) while accommodating for interval censored data may facilitate early detection of MIC creep. As a result, these methods are able to detect effects of parameters that may be missed by logistic regression, making them a useful tool for surveillance [34]. However, accelerated failure time models and related approaches are less robust to heavy left or right censoring in the data than logistic regression and may be more complex to implement for those with less experience in modeling [34]. Accelerated failure times also may encounter problems estimating variance for regression parameters and the possibility of finding multiple sets of optimal parameter values [74].

An overview of the various models presented, including a summary of their advantages and disadvantages, can be found in Table 1.

## 5. Conclusions

MICs provide an important tool for surveillance of phenotypic resistance, allowing for assessment of trends in AMR when analyzed with appropriate modeling approaches. They represent a long-standing method for assessment of levels of resistance of bacterial populations that remain widely used in clinical and surveillance settings. As the analysis of AMR shifts toward WGS-based analysis techniques it is important to consider the utility of MICs and other phenotypic approaches. While the nature of MIC data makes its analysis complex and non-uniform, MICs provide valuable and unique insights on resistance patterns, including adaptive resistance, that can be paired with WGS and metagenomics to provide more insight into acquired resistance. As we find out more about bacterial community structure, understanding and measuring phenotypic changes via MIC based approaches will continue to provide important insights as the world confronts the issue of AMR. 

The analysis of MIC data requires a tradeoff between model simplicity and information loss, as simpler models such as logistic regression lose information conveyed by the original MIC value by simplifying the dependent variable. However, they also allow for easier communication with a general audience and may more easily accommodate data from disparate sources (e.g., multiple laboratories using different MIC levels). More complex models retain more of the information from MICs but may be less appropriate when there are changes in ranges of concentrations tested, if there is heavy left or right censoring (mixture models or AFT models), or if the distribution of WT and non-WT distributions heavily overlap (mixture models).

As a result, the choice depends heavily on the purpose of the analysis. For surveillance programs it may be advantageous to detect any changes in resistance patterns on the whole continuous MIC scale with a mixture model, but research comparing changes in prevalence of MICs classified as resistance phenotypes may only require logistic regression. In some instances, the critical question of a study may be: what is the prevalence of bacterial isolates with MICs that exceed a threshold of interest, such as the clinical classification of “resistant”? In these instances, a binary outcome variable simplifies the regression approach and the interpretability of regression coefficients, while still addressing the objective of the study. One should also recognize that cutoffs for ECOFF and clinical breakpoints are carefully selected to be as informative as possible, and that clinical breakpoints are set to highlight specific changes in resistance level that are relevant to expected effectiveness of treatment of humans with antimicrobials in clinical settings. It is also important to note that these cutoffs still only represent a single point, and as such cannot account for the variability in effectiveness of administration in all settings, in this respect broader knowledge about the full distribution of MICs may be an important tool to determining effective courses of treatment in addition to their utility for surveillance.

It is important to acknowledge that there may be no perfect model for every situation. The selection of a modeling approach must be made with consideration of a number of factors, including the context of the data, ease of communication of the model, desired results, and historical precedent.

## Figures and Tables

**Figure 1 animals-10-01405-f001:**
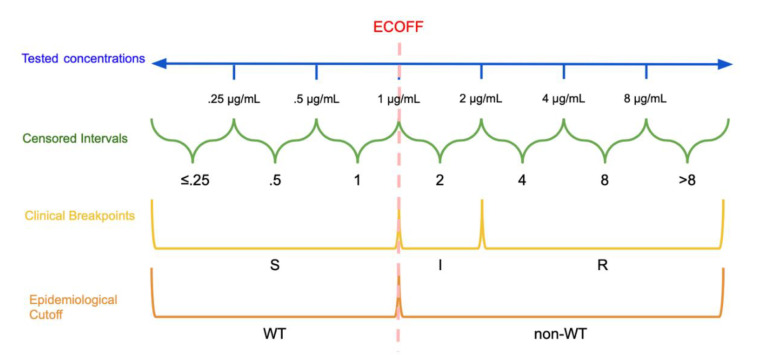
Dichotomization and Categorization of Minimum Inhibitory Concentration (MIC) data: In this hypothetical example, the continuous scale of concentrations is divided into intervals based on the tested concentrations. The epidemiological cutoff (ECOFF) divides wild type (WT) and non-wild type isolates at 1 μg/mL while the clinical breakpoints divide the MICs into susceptible (S): ≤1 μg/mL, susceptible, increased exposure (I): >1 μg/mL and ≤2 μg/mL, and resistant (R): >2 μg/mL. Note that the ECOFF and S breakpoint need not be the same.

**Table 1 animals-10-01405-t001:** Summary of Advantages and Disadvantages of MIC Modeling Approaches.

Model	Data Type	Advantages	Disadvantages
Logistic regression	Dichotomous	Identifies changes in proportions of MICs above and below cutoffs and their relationship to covariates Dichotomization ignores effects of heavy left and right-censoring More suitable than other methods for data with differences in ranges of concentrations tested as long as cutoff for dichotomization is included in the scales Easy interpretation and communication	MIC Creep may occur due to changes in MIC distributions that do not cross the cutoff Does not provide information on the density distribution of the full scale of MICs (e.g., cannot find the distribution of MICs among non-WT bacteria)
Proportional odds cumulative logit model	Ordinal	Avoids reliance on a single cutoff Observing effects of covariates if proportionality assumption is met Easy interpretation of coefficients related to logistic regression	Not appropriate for data with different ranges of concentrations tested Proportionality odds assumption must be satisfied to fit model Does not provide insight into MIC probability distribution
Generalized ordered logit model	Ordinal	Avoids reliance on a single cutoff Observe effects of covariates with no requirement of proportionality Easy interpretation of coefficients related to logistic regression	Not appropriate for data with different ranges of concentrations tested Does not provide insight into MIC probability distribution
Mixture models	Interval-Censored	Finds distributions of WT and non-WT bacteria as well as appropriate cutoff values Can identify changes in distributions for different levels of covariates Estimates of proportions of WT and non-WT bacteria.	Some methods are not appropriate for data with different ranges of concentrations tested Ability to distinguish between subpopulations depends on amount of separation May not be appropriate for distributions with heavily left- and right-censored data because mixture models cannot accurately fit distributions on data known only to lie above or below the scale
Accelerated failure time model with frailties	Interval-Censored	May avoid reliance on a single cutoff Estimates effects of covariates on expected concentration to inhibition of growth	Not appropriate for heavily left- and right-censored data Numerical challenges including the possibility of multiple solutions to the estimating equations, and difficulty in estimating variance of the regression parameters

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
