# Peer review of "Overview of Quantitative Methodologies to Understand Antimicrobial Resistance via Minimum Inhibitory Concentration"

_animals, 2020, doi:10.3390/ani10081405_

Round 1

Reviewer 1 Report

The paper by Michael et al. explores statistical methods for analyzing MIC data. This is an important topic, given that many laboratories continue to generate data using standard antimicrobial susceptibility testing methodologies. The challenge has always been how to analyze these data. This paper very nicely describes a range of approaches for MIC data analysis, including a delineation of strengths and weaknesses of different approaches. Table 1, for example, is a very useful synthesis of this material. Detailed below are some minor issues that should be addressed in a revision of the manuscript.

The paper uses ECOFFs extensively in the paper. One problem with ECOFFs is that the breakpoints can be arbitrary and regionally specific, whereas breakpoints for MIC are supposed to be based on actual ability to get the drug to target location. Regardless, the paragraph that starts on line 121 is worded in a way that compounds one of the fundamental misunderstandings of ECOFFs, namely that non-WT has nothing to do with resistance. The authors do a good job of explaining this elsewhere in the paper, including in Figure 1, but because this paragraph introduces the concept, the authors need to bring this discussion to the forefront. For example, the authors write “separates wild-type (WT/non-resistant) organisms from those that have acquired phenotypic resistance mechanisms (non-WT/resistant).” Please remove this notation, as non-WT are not necessarily resistant and make sure that this paragraph explicitly makes this point (and echoes it throughout the paper).

Section 2.2 beginning on Line 162 requires the dichotomization of the MIC data. However, the authors never really define what the two mutually exclusive categories are. For MIC data with clinical breakpoints, is the comparison SI versus R or S versus IR? I would hope that the authors could make this point explicitly that the only correct way to analyze these data is to put I with S (and thus compare SI versus R). The Intermediate MIC isolates are NOT resistant and should never be analyzed with R.

The paragraph beginning on Line 235 does a good job describing the method, but I wonder whether it actually makes any sense from a biological perspective. The authors have already spent time describing MIC creep, but this is often a microbial population-based phenomenon and not one seen in individual isolates. When bacterial isolates acquire a new resistance mechanism, they often make very large jumps in MIC (not creep). Perhaps stepwise mutational events through drift would result in creep, but gene acquisition typically results in large MIC changes. For this reason, I wonder whether the assumptions of this approach would ever be met. This ties into my main point below, namely the biological aspects of these statistical approaches.

The paragraph on Accelerated Failure Time Models could use some editing. As written, it is hard to understand how a time failure model is applied to concentrations of antimicrobials. Additional explanation would be helpful. The second paragraph, specifically lines 405-406, provide some of this, but this needs to be moved up and expanded.

Another issue that is not really addressed in this paper with respect to interval-censored data is the fact that as antimicrobial concentration increases, the size (width) of the interval increases. Of course, a log2 transformation eliminates this, but for the AFT models and others, how does the model account for the fact that the interval had higher concentrations can be massive in terms of ug/ml antimicrobial?

Overall, this paper is very interesting and useful, but my biggest concern is that the link of these methods to actual studies that have used the methods is very weak. I would have liked to see a paragraph in each section showing how authors have applied these methods. As it is written, there are few references for methods with a sentence describing a study that used the method, but how was the method used? Which covariates were included? Is there a study design specific to each method? Without the context, the paper is very dense. I am not sure if the authors have room to add biological examples to each section, but it might improve readability and would definitely give some added perspective.

Finally, remember that lab variation can be result in plus or minus one dilution.

Other concerns:

Line 55: This paragraph mentions the NARMS program. This is a U.S. program, and because readership is international, all references to programs like NARMS should be preceded with U.S., such as “The U.S. National Antimicrobial…”

Line 61: The sentence starts with “The MIC is determined” but later the sentence mentions disk diffusion. There is no MIC generated by a disk diffusion.

Line 76: the sentence says “reported as “≤J μg/mL” where J is the highest concentration tested.” I think this is supposed to be the lowest concentration tested.

Line 225: Wouldn’t the ordinal data retain the two-fold relationship if it were analyzed on a log2 scale?

Line 310: delete “the” after “distribution”

Line 471: Should be “levels”

Table 1: in Generalized ordered logit, bullets 2 and 3 in Advantages are probably supposed to be together.

Reviewer 2 Report

The reviewer found this a fascinating piece of work. The authors made some good arguments and supported it with good references. The manuscript will definitely caused much debate and I really want to comment the authors for this work. What I was not sure about is if the theme of this manuscript fit the scope of this journal much focused on animal themes. This work could very well fit in antimicrobial research, antimicrobial therapy or drug development journals as well.  However, will leave that for the editors to decide.  

Just minor comments

The abstract does not conclude the authors outcome of the review and appear incomplete. Another line is needed here.  

Line 310: Remove second 'the'

Author Response

Thank you very much for your input. We restructured the second half of the abstract to focus more on the goals and conclusions of the study, emphasizing some of the key considerations for model selection.

Also deleted the second "the" in line 310.